# Combined Pseudo-Random Sequence Generator for Cybersecurity

**DOI:** 10.3390/s22249700

**Published:** 2022-12-11

**Authors:** Volodymyr Maksymovych, Mariia Shabatura, Oleh Harasymchuk, Ruslan Shevchuk, Pawel Sawicki, Tomasz Zajac

**Affiliations:** 1Department of Information Technology Security, Lviv Polytechnic National University, 79013 Lviv, Ukraine; 2Department of Information Security, Lviv Polytechnic National University, 79013 Lviv, Ukraine; 3Department of Computer Science and Automatics, University of Bielsko-Biala, 43-309 Bielsko-Biala, Poland; 4Department of Computer Science, West Ukrainian National University, 46009 Ternopil, Ukraine; 5ITSO GmbH, D-10829 Berlin, Germany

**Keywords:** pseudo-random number, pseudo-random sequence generators, authentication, encryption of information

## Abstract

Random and pseudo-random number and bit sequence generators with a uniform distribution law are the most widespread and in demand in the market of pseudo-random generators. Depending on the specific field of application, the requirements for their implementation and the quality of the generator’s output sequence change. In this article, we have optimized the structures of the classical additive Fibonacci generator and the modified additive Fibonacci generator when they work together. The ranges of initial settings of structural elements (seed) of these generators have been determined, which guarantee acceptable statistical characteristics of the output pseudo-random sequence, significantly expanding the scope of their possible application, including cybersecurity. When studying the statistical characteristics of the modified additive Fibonacci generator, it was found that they significantly depend on the signal from the output of the logic circuit entering the structure. It is proved that acceptable statistical characteristics of the modified additive Fibonacci generator, and the combined generator realized on its basis, are provided at odd values of the module of the recurrent equation describing the work of such generator. The output signal of the combined generator has acceptable characteristics for a wide range of values of the initial settings for the modified additive Fibonacci generator and the classic additive Fibonacci generator. Regarding the use of information security, it is worth noting the fact that for modern encryption and security programs, generators of random numbers and bit sequences and approaches to their construction are crucial and critical.

## 1. Introduction

Pseudo-random number (PRNG) and bit sequence (PRBSG) generators, in general, pseudo-random sequence generators (PRSG), are used in many fields of science and technology [1,2,3,4,5,6,7]. They play an important role in modeling various processes, in solving industrial problems and cybersecurity problems. The relevance of cybersecurity tasks is growing every year. Such tasks include encryption of information, generation of secret keys, authentication, confidentiality and integrity of information. Additionally, an effective solution to these problems is not possible without the use of PRNG and PRBSG.

Random numbers are an important input for many Internets of Things (IoT) functions [8]. In [9], a deep review is given for lightweight random and pseudo-random number generators designed for constrained devices, such as wireless sensor networks and RFID (Radio Frequency Identification) tags, along with a study of a Trifork pseudo-random number generator for constrained devices. In [10], the authors focus on the security and privacy risks in cloud databases and provide a solution for clients who want to generate the pseudo-random number collaboratively in a distributed way which can be reasonably secure, fast and low-cost to meet the requirements of a cloud database. In [11], the use of evolutionary computation is proposed for designing and optimizing lightweight PRNGs. In [12], the authors introduce a novel approach to implement Pseudo Random Number Generators, by proposing the use of generative adversarial networks (GAN) to train a neural network to behave as a PRNG.

In [13], a text algorithm for watermarks based on a PRNG for use in cryptography was proposed, which has good invisibility and reliability to withstand removal, modification attacks, etc., and can be used in the field of hiding information using cloud computing. An interesting method is proposed in [14], whereby random numbers are generated based on reinforcement learning characteristics that select the optimal behavior considering every possible status up to the point of episode closing to secure the randomness of such random numbers.

A number of questions about the approaches to the construction of PRNG and PRBSG, as well as the requirements for their use in cybersecurity systems are discussed in many works, where the authors explore known methods of constructing such generators, analyze their weaknesses and search for new generation methods [15,16,17,18,19,20,21,22,23,24].

Among PRNG, it is possible to allocate additive Fibonacci generators (AFG) to a separate group [25,26,27,28,29,30,31,32,33], which is characterized by the fact that the sequences at their output have acceptable statistical characteristics. This group of generators is effectively used for mathematical and statistical calculations, as well as in the modeling of various processes. It should be noted that AFG, in contrast to linear congruent generators, can be used in algorithms that are critical to the quality of generated pseudo-random numbers, particular in cryptography. The ever-increasing possibilities for the use of AFG and the new requirements for them pose new challenges to developers in their design and implementation; therefore, in this direction, they are actively working to find new structures of modified additive Fibonacci generators (MAFG) with improved characteristics. In particular [34], the authors propose a five-state real-time generator for the Fibonacci sequence and give formal proof of the correctness of the generator. The proposed five-state Fibonacci sequence generator is optimum in generation steps and is implemented on the smallest known finite state automaton in the number of states. In [35], the researchers sought to determine, between the Fibonacci Random Number Generator and the Gaussian Random Generator, which is better for improving data security in cryptographic software systems.

To assess the quality of PRNG and PRBSG, it is necessary to use statistical testing based on various randomization algorithms and obtain an integral picture of the assessment of the pseudo-random sequence at the generator output. The National Institute of Standards and Technology (NIST), which is part of the American National Standards Institute (ANSI), the International Organization for Standardization (ISO), and national standardization authorities are developing assessment methods, requirements, and standards for PRNG and PRBSG. Among the most popular tests for quality assessment is a set of statistical tests NIST [36,37,38,39,40]. If the generated sequences are aimed to be used in cryptographic applications, it is also necessary to conduct cryptographic testing.

In general, an extremely large number of specialists are involved in the design, construction, quality assessment and application of PRNG and PRBSG.

In comparison with the known ones, the novelty of this work lies in the optimization of the parameters of the MAFG, which includes the possibility of realizing an arbitrary value of the modulus of the recurrent equation of the generated polynomial. Additionally, further improvement was achieved in the output signal characteristics of the pseudo-random generator through the joint work of the MAFG and the classical AFG.

### 1.1. Related Literature

In [40,41,42,43,44,45,46,47,48,49], we and another scientist proposed MAFG, which differs from that reported in [25,26,27,28,29,30,31,32,33,34,35,36] as follows:-The possibility of functioning according to recurrent equations, the modules of which can have arbitrary values;-The presence of an additional logic circuit that can significantly improve the statistical characteristics of the output signals;-The presence of structures and relevant hardware models, which allows them to be implemented in modern element bases.

Advantages of the new MAFG are demonstrated by examples of their work in accordance with several recurrent equations [40,41,42,43,44,45].

In connection with the emergence of our new MAFG, a number of questions arise, which include the following:-The choice of polynomials on the basis of which it is possible to synthesize MAFG with acceptable statistical characteristics;-Determination of the generator’s structural element parameters at which acceptable statistical characteristics are reached;-The method of joint work of the MAFG with classical AFG in order to further improve the statistical characteristics of the output sequence and increase cryptosecurity;-The definition and expansion of ranges of structural elements initial installations (seed) at which acceptable statistical characteristics of the output pseudo-random sequence are guaranteed;-The search for new approaches to the design of the MAFG and methods of their implementation, in particular on the Programmable Logic Device;-The selection of methods for testing the quality of sequences obtained from such generators for compliance with given conditions.

Table 1 shows a summary of the comparison between the research literature and the work developed in this study. If a column item is marked, the work in that row addresses it. If it is not marked, then the work either does not specify or does not address that item. The items of the table are as follows:
Work—contains a reference to research;Repetition period—if the work investigated the repetition period of the generator’s output sequence;Statistical characteristics—if the work investigated and showed statistical characteristics of the output sequence of the generator;Ranges of key—if the work investigated the value ranges of key information;Polynomial—if it was possible to use an arbitrary value of the modulus of the recurrent equation of the polynomials forming the algorithm of the generator implemented;Compatible work—if the compatible work of several generators was possible;Hardware implementation—if the work showed the hardware implementation of the generator, in addition to software.

### 1.2. Purpose of Work

The aim of the work is to optimize the structures of classical AFG and MAFG in their joint work, to determine the ranges of initial settings of structural elements (seed) AFG and MAFG, which guarantee acceptable statistical characteristics of the output pseudo-random sequence.

## 2. Materials and Methods

### Structure Scheme of the Joint Work of AFG and MAFG and the Principle of Its Work

The generalized scheme of joint work of AFG and MAFG is given in Figure 1, and the corresponding detailed structure scheme is given in Figure 2.

The output pseudo-random bit streams AFG—bi and MAFG—di are combined through the logical element XOR, thus forming the output bit sequence—bdi. The inputs AFG and MAFG receive clock pulses—$f_{c}$. Before starting the work, the initial values (seed) are written to the AFG and MAFG memory registers X_0 and Y_0 accordingly.

In this paper, the characteristics of the device are investigated, despite the fact that AFG is constructed in accordance with the primitive polynomial
(1)$$GF\left(2^{30}\right)=x^{30}+x^{23}+x^{2}+x+1,$$
and MAFG is constructed according to the primitive polynomial
(2)$$GF\left(2^{20}\right)=y^{20}+x^{3}+1,$$

The choice of polynomials (1) and (2) was made on the basis of studies of many other variants and was based on achieving acceptable statistical characteristics of the output pseudo-random signal, in the full range of possible values X_0 and Y_0 with minimal hardware costs.

AFG consists of adders $AD_{1}-AD_{3}$ and memory registers $RG_{1}-RG_{30}$ (Figure 2).

According to polynomial (1), AFG operates on a recurrent equation
(3)$$x_{i}=\left(x_{i-30}+x_{i-23}+x_{i-2}+x_{i-1}\right)\mathrm{mod}m,$$
where $x_{i}$ is the adder $AD_{3}$ output number; $x_{i-1}$, $x_{i-2}$, $x_{i-23}$, $x_{i-30}$ are numbers at register $RG_{1}$, $RG_{2}$, $RG_{23}$, $RG_{30}$ outputs, respectively; and m is the modulus of the recurrent equation, determined, in this case, by the number of binary bits of the structural elements of AFG.

The output bit sequence bi is formed at the output of the least significant bit of the adder $AD_{3}$.

The MAFG consists of adders $AD_{1}$ and $AD_{2}$, multiplexer MUX, memory registers $RG_{1}-RG_{20}$, logic element OR and logic circuit LC.

In accordance with polynomial (2) and the new method of its internal construction [42], MAFG operates on a recurrent equation
(4)$$y_{i}=(y_{i-20}+y_{i-3}+a)\mathrm{mod}h,$$
where $y_{i}$ is the multiplexer MUX output number; $y_{i-3}$ and $y_{i-20}$ are register $RG_{3}$ and $RG_{20}$ outputs numbers; h is the the module of the recurrent equation, which is determined, in this case, by the number of binary bits of the structural elements of the MAFG and the value of the control code A that is coming to the second input of the combination adder $AD_{2}$; and a is the output signal of the logic circuit LC, which is determined by the equation
(5)$$a=a_{0}⊕a_{1}⊕\cdot \cdot \cdot ⊕a_{s},$$
where $a_{i}$ (i=0,1,…s;s≤n−1) are the bit values of the number $y_{i-1}$, and n is the number of its binary bits.

It should be emphasized that, due to its original structure, the MAFG can operate in accordance with a recurrent equation with an arbitrary value of the module, which will be used in further studies of this work.

The output bit sequence di is formed at the output of the multiplexer MUX’s least significant bit.

## 3. Results and Discussion

### 3.1. Results of AFG Research

The main task of AFG, which works in pair with MAFG, is to ensure a stable repetition period of output pseudo-random sequence on the whole set of initial values X_0 in registers that are part of it.

It is known [36] that the maximum period of the AFG output sequence is achieved if it works following the primitive polynomial.

As mentioned earlier in this paper, the selected classical AFG works following the primitive polynomial (1) and, therefore, the recurrent Equation (3).

At m=2, the repetition period of the AFG output signal is equal to $2^{30}-1$ for any initial value X_0 [43].

Figure 3 shows the statistical characteristics of the AFG output signal obtained using statistical tests NIST [37,38,39]. The results are presented in the statistical portraits. A statistical portrait is an mq matrix, where m is the number of binary sequences to be tested and q is the number of statistical tests [37].

According to the obtained statistical portrait, the proportion of sequences that passed each statistical test is determined. For this, setting the significance level α and the probability values p exceeding the established level α are calculated for each q tests.

It is considered that the generator has passed the test if the value of the coefficient is within $\left[r_{\max},r_{\min}\right]$. The confidence interval limits are determined by the following expression [37]:(6)$$r_{\max(\min)}=p\pm 3\sqrt{\frac{p(1-\alpha )}{m}},$$
where p=1−α.

By substituting the corresponding values, namely α=0.01 and m=1000, we obtain the limits of the confidence interval 0.980561−0.999439. A sequence is said to meet randomness requirements when all test values fall within these limits.

The NIST statistical test set consists of 15 tests, calculating 188 results values [37]. Each of these 188 values is interpreted as the result of a separate test. Each of these 188 values is marked as a dot in the statistical portrait.

The statistical portrait is obtained at the initial value of numbers X(0)=7. Similar statistical portraits were obtained for other values X(0). The output AFG sequence does not pass all the tests from the NIST set, so such a generator is not statistically safe.

### 3.2. Results of MAFG Research

We will research the characteristics of MAFG for the case of its operation in accordance with the primitive polynomial (2).

Figure 4 and Figure 5 show the dependences of the repetition periods *T* on the initial values of numbers Y_0 in the registers $RG_{1}-RG_{20}$. The value Y_0 is determined by the equation:(7)$$Y\_0=h^{19}y_{i-1}(0)+h^{18}y_{i-2}(0)+h^{17}y_{i-3}(0)+\ldots +hy_{i-19}(0)+y_{i-20}(0),$$
where $y_{i-1}(0)$, $y_{i-2}\left(0\right)$, …, $y_{i-20}\left(0\right)$ are the initial values of the numbers in registers $RG_{1},RG_{2},\;\ldots ,RG_{20}$.

Dependencies are obtained only for two h values and one value of the LC output signal a because obtaining such dependencies for other values of these parameters requires a significant increase in computing resources and machine time.

Here, it should be emphasized that the constancy of the MAFG output sequence repetition period, for the whole set of the initial values Y_0, is guaranteed only at h=2 and a=0.

Investigations have shown that for other h and a values, the repetition period significantly varies depending on Y_0, achieving under specific Y_0 small values, thus generating so-called “weak keys”.

This circumstance is the main reason that leads to further research, provided that the MAFG works in conjunction with the classic AFG, which ensure acceptable minimum values of the period for any values Y_0.

Analyses of the statistical characteristics of the MAFG output signal allowed us to make the following conclusions.

Statistical characteristics at even values of h are much better than at odd values. This is due to a certain asymmetry in the formation of the MAFG output signal di at odd values of h.

In confirmation of this, Figure 6 shows statistical portraits of the MAFG output signal for Y_0=7 Y.

The graphs show statistical portraits of two generators. Figure 6a shows that such a sequence does not meet the randomness requirements since it passed only 2 out of 15 tests (Cumulative Sums and Serial) (see Table 2) of the NIST set. Figure 6b shows that all values are within the confidence intervals; therefore, such a sequence meet the randomness requirements. Table 2 shows the results of the testing calculations.

It was also found that the statistical characteristics of the MAFG output signal improve when forming the output signal of the LC following the logic equation
(8)$$a=a_{0}⊕a_{r}$$
where $a_{0}$ and $a_{r}$ are the values of the least and most significant bits of register RG1 in the MAFG structure.

In confirmation of this, Figure 7 and Figure 8 shows statistical portraits of the MAFG output signal for h=8, variable initial values Y_0=7,32,100 and different values of the LC output signal: $a=a_{0}⊕a_{1}$ (Figure 7) and $a=a_{0}⊕a_{2}$ (Figure 8).

Figure 7 shows that by using such initial values, the generator formed a non-random sequence. There are always one or two tests that fail the NIST tests.

Finally, we succeeded in passing all NIST tests at values of 32 and 100. Comparing Figure 7 and Figure 8, we conclude that the quality of the pseudo-random sequence is affected by the value of the output signal of the LS−a.

Besides investigating the statistical characteristics of the MAFG, we determined the repetition periods. Table 3 shows some results of the investigation of the MAFG repetition period.

Therefore, as a result of the research on MAFG statistical characteristics, it is established that they essentially depend on values h and a.

### 3.3. Results of Research Combined Steam Generator AFG and MAFG

This section presents the results of the research output sequence of pair AFG and MAFG (Figure 1 and Figure 2), with a proven assumption that the improvement of the MAFG statistical characteristics for any initial value Y_0 can be achieved by forming the output signal of the logic circuit LC in such way that the least and most significant bits of the register $RG_{1}$ are fed to the input of the LC.

Table 4 shows the parameters of the logic circuit LC following logic Equation (4) for different values *h*, in which the statistical characteristics of the output signal of the combined generator bdi (Figure 1 and Figure 2) were studied at different initial values X_0 and Y_0. Figure 9 presents the results of these studies.

The obtained results (Figure 9) confirm the assumption that at even values of *h* and upon the formation of the LC output signal following Equation (4), the output signal of the combined generator successfully passes all 188 tests of the NIST set at different initial values of structural elements AFG and MAFG, which ensure its statistical security as one of the main components of cryptographic stability.

Figure 9 shows the results of testing the combined generator at *h* = 10.

Table 5 shows the results of passing the combined generator at h=10, $a=a_{0}⊕a_{3}$ with initial values X_0=7,Y_0=7 (generator 1), X_0=7,Y_0=32 (generator 2), X_0=7,Y_0=100 (generator 3).

Figure 10 shows the summary results of testing the combined generator with the parameters displayed in Table 5.

Figure 10 shows that combined generators with an even value of h meet the randomness requirements according to the NIST method in a wide range of initial values X_0 and Y_0, and with an odd value, they do not. We decided to investigate this issue in more detail.

Further research proved that acceptable statistical characteristics of the output signal of the combined generator could also be provided with odd values of h, but in narrower ranges of initial values of X_0 and Y_0. This is confirmed by the statistical portraits in Figure 11.

As can be seen from Figure 10, the combined generator with h=9 has the worst statistical characteristics. By doing additional research, such as varying the initial numbers in X_0 and Y_0 with the condition that $a=a_{0}⊕a_{3}$ we managed to achieve the passing of all NIST tests.

The results of the tests are shown in Figure 11, where the left side shows the combined generator with initial values X_0=7, and variables Y_0=36 (Figure 11a), 64 (Figure 11c), 71 (Figure 11e), 100 (Figure 11g), 1004 (Figure 11i). On the right side is the combined generator with X_0=32 and variables Y_0=36 (Figure 11b), 64 (Figure 11d), 71 (Figure 11f), 100 (Figure 11h), 1004 (Figure 11j). The value of the output signal of the LS for both cases is the same $a=a_{0}⊕a_{3}$.

The statistical portrait Figure 11 shows that the combined generator at X_0=7, Y_0=32 did not pass four NIST tests. This means that the sequence does not meet the randomness requirements. After changing the value of X_0=32, we see an improvement, and all tests are passed (Figure 11c). The situation is similar to the combined generator Y_0=100 (Figure 11g,h). Summarizing, we can conclude that the change of the initial numbers X_0 and Y_0 affects the quality of the statistical characteristics of the combined generator.

Thus, as a result of experimental research, we managed to choose the parameters of the combined generator so that it allows the formation of a sequence with acceptable statistical characteristics for odd *h*.

## 4. Conclusions

This work offers a new structure of the combined generator consisting of classical AFG and modified AFG−MAFG. The output pulse sequence of the combined generator is formed as the sum modulo 2 of the output sequences of AFG and MAFG.

At certain values of the AFG, internal parameters provide the maximum value of the repetition period of the output sequence for all possible initial settings X_0 of structural elements, thereby providing a repetition period of the combined generator output signal not less than a certain specified value.

It is proved that the statistical characteristics of the MAFG output signal and the combined generator are generally acceptable at odd values of the parameter *h*, determined by the number of bits of the MAFG registers, and the method of constructing a logic circuit, at which input signals come from the most and least significant bits of one of the MAFG registers. The statistical characteristics of the combined generator output signal are acceptable for a wide range of values of the initial settings AFG−X_0 and MAFG−Y_0.

The scientific novelty of this paper lies in the further optimization of the MAFG structure with an arbitrary value of the modulus of the recurrent equation, the working principle of which was proposed by the authors in previous works, and the investigation of compatible work with the classical AFG. It allows for significantly improved basic characteristics of additive Fibonacci generators. An essential factor is that the development is aimed, first of all, at the hardware implementation, which allows the high-speed generators to be ensured.

The purpose of further research in this direction will be practical recommendations for the choice of parameters of the combined generator depending on the specific tasks of creating cryptographic or other technical means.

One possible direction for further research in this direction may be to investigate structures with more of AFG and MAFG combined in a different way.

## Figures and Tables

**Figure 1 sensors-22-09700-f001:**
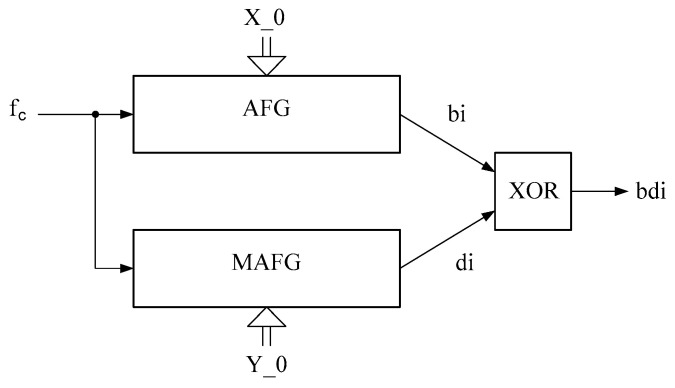
Generalized scheme of the joint work of AFG and MAFG.

**Figure 2 sensors-22-09700-f002:**
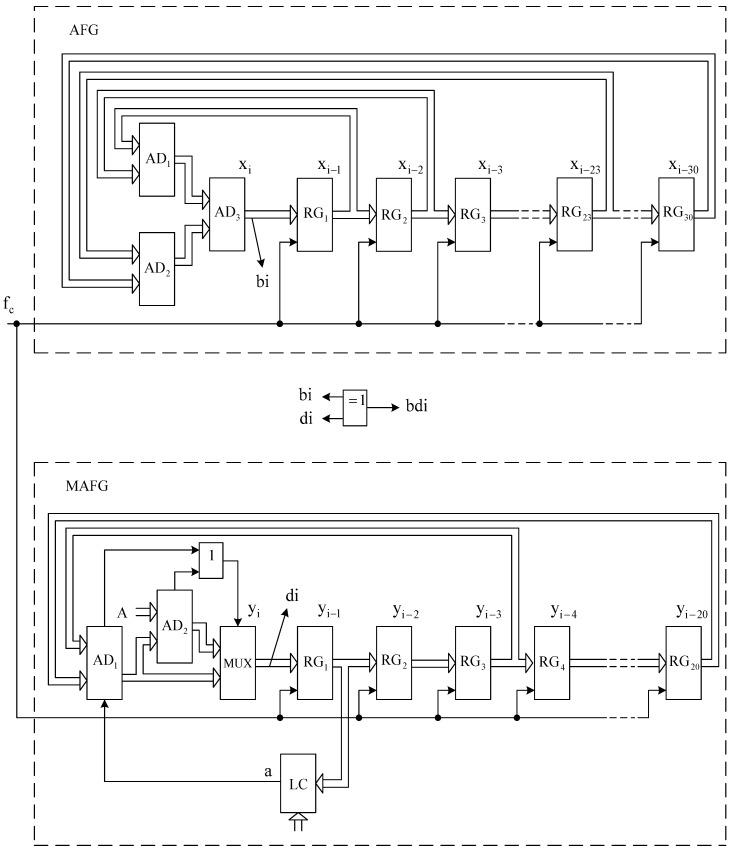
Structure scheme of the joint work of AFG and MAFG.

**Figure 3 sensors-22-09700-f003:**
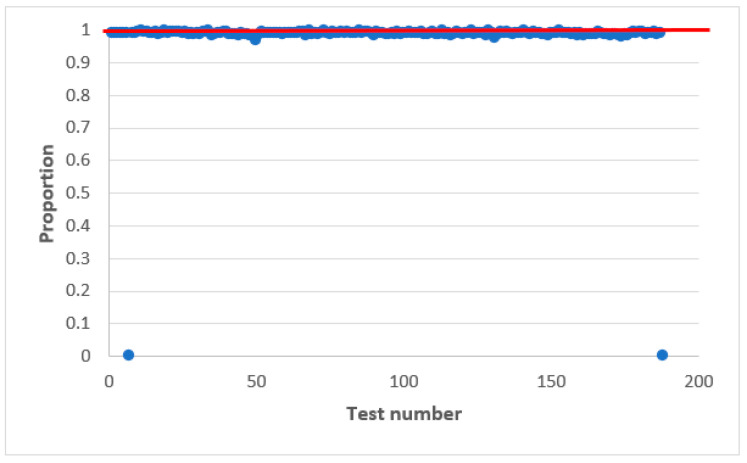
Statistical portrait of the AFG output sequence.

**Figure 4 sensors-22-09700-f004:**
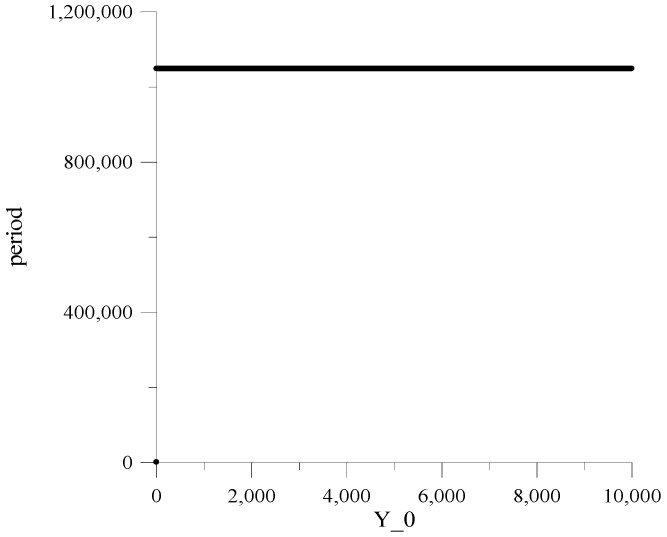
Dependencies of repetition periods for X_0, h=2 and a=0.

**Figure 5 sensors-22-09700-f005:**
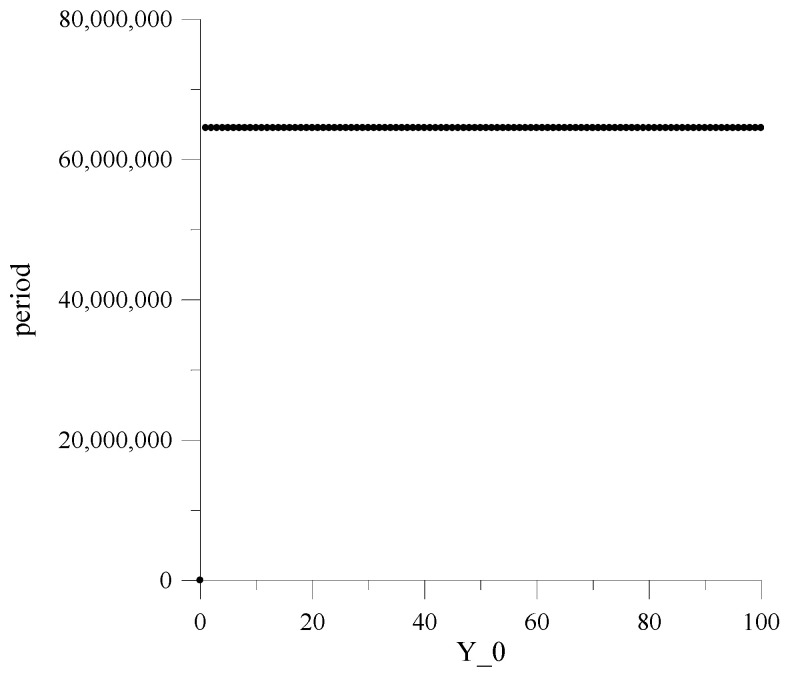
Dependencies of repetition periods for X_0, h=3 and a=0.

**Figure 6 sensors-22-09700-f006:**
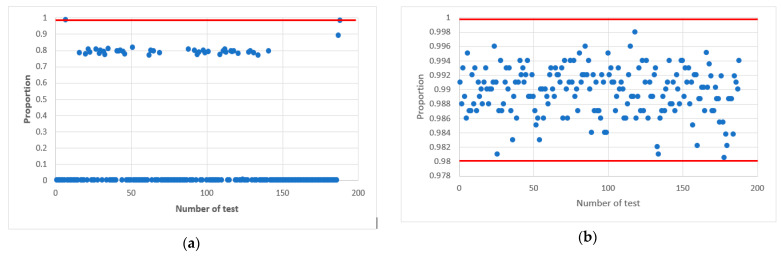
Statistical portrait of the MAFG output sequence: (**a**) with an odd value h=9, (**b**) at an even value h=10.

**Figure 7 sensors-22-09700-f007:**
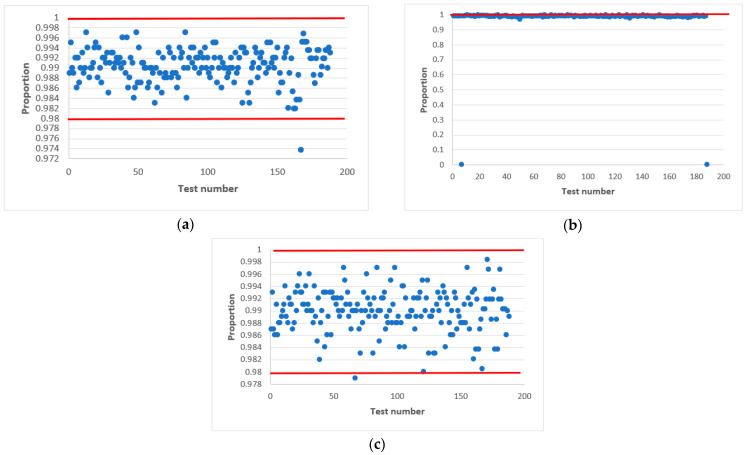
Statistical portrait of the MAFG output sequence for h=8, $a=a_{0}⊕a_{1}$ with initial values: (**a**) Y_0=7, (**b**) Y_0=32, (**c**) Y_0=100.

**Figure 8 sensors-22-09700-f008:**
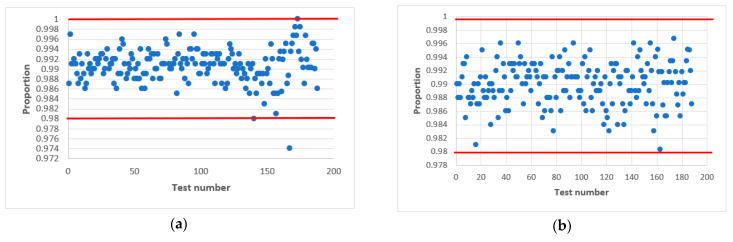
Statistical portrait of the MAFG output sequence for h=8, $a=a_{0}⊕a_{2}$ with initial values: (**a**) Y_0=7, (**b**) Y_0=32, (**c**) Y_0=100.

**Figure 9 sensors-22-09700-f009:**
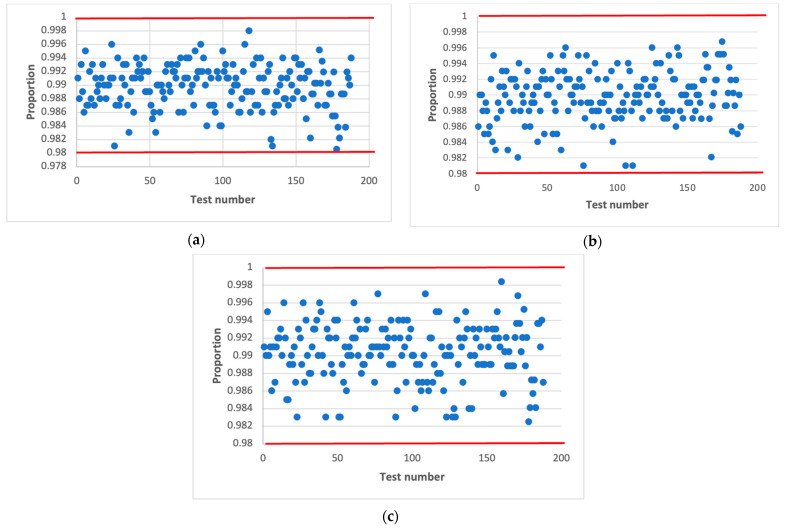
Statistical portrait of the combined generator output sequence for *h* = 10, $a=a_{0}⊕a_{3}$ with initial values: (**a**) Y_0=7, (**b**) Y_0=32, (**c**) Y_0=100.

**Figure 10 sensors-22-09700-f010:**
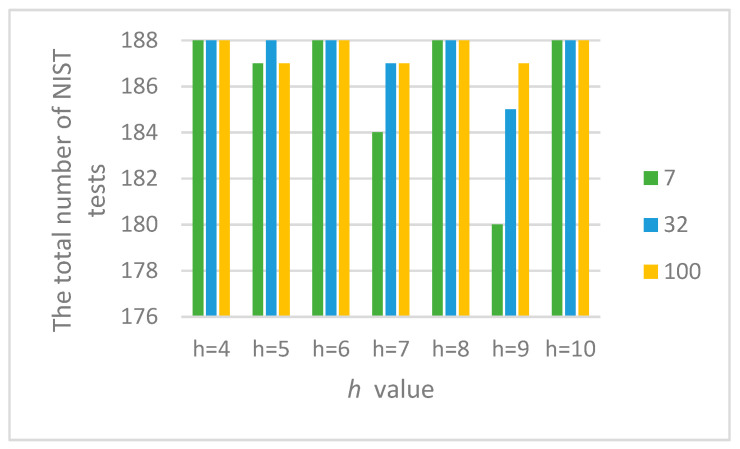
Results of passing the NIST tests with the combined generator.

**Figure 11 sensors-22-09700-f011:**
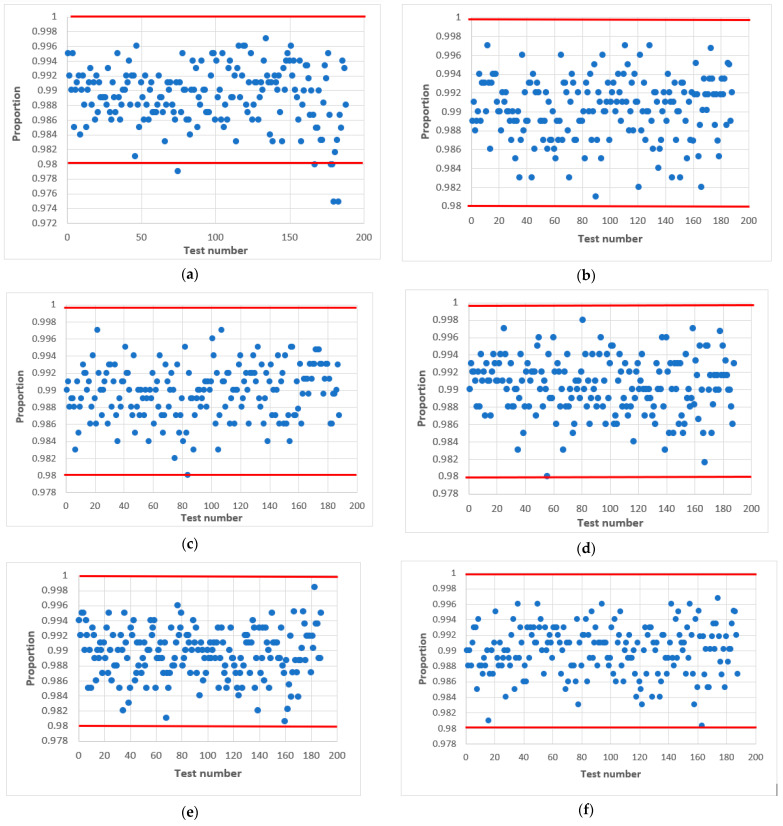
Statistical portrait of the combined generator for *h* = 9, $a=a_{0}⊕a_{3}$ with initial values: (**a**) X_0=7,Y_0=36; (**b**) X_0=32,Y_0=36; (**c**) X_0=7,Y_0=64; (**d**) X_0=32,Y_0=64; (**e**) X_0=7,Y_0=71; (**f**) X_0=32,Y_0=71; (**g**) X_0=7,Y_0=100; (**h**) X_0=32,Y_0=100; (**i**) X_0=7,Y_0=1004; (**j**) X_0=32,Y_0=1004.

**Table 1 sensors-22-09700-t001:** Comparison of related work with the study in this paper.

Work	Repetition Period	Statistical Characteristics	Ranges of Key	Polynomial	Compatible Work	Hardware Implementation
[2]		X				X
[3]	X	X	X			
[8]		X				
[11]						X
[25]		X			X	
[27]		X				
[28]	X	X	X			
[30]		X	X			X
[31]	X					X
[33]	X	X	X		X	
[35]	X	X	X			
This work	X	X	X	X	X	X

**Table 2 sensors-22-09700-t002:** Results of testing MAFG by NIST.

№	Statistical Test	Figure 6a		Figure 6b	
		*p*-Value	Status	*p*-Value	Status
1	Frequency	0.000000	-	0.291091	+
2	Block Frequency	0.000000	-	0.664168	+
3	Cumulative Sums	0.576961	+	0.950247	+
4	Runs	0.000120	-	0.377007	+
5	Longest Run	0.000750	-	0.350485	+
6	Rank	0.000106	-	0.757790	+
7	FFT	0.000132	-	0.192724	+
8	Non-Overlapping Template	0.000000	-	0.626709	+
9	Overlapping Template	0.000000	-	0.368587	+
10	Universal	0.000000	-	0.289667	+
11	Approximate Entropy	0.000000	-	0.096000	+
12	Random Excursions	0.000000	-	0.958867	+
13	Random Excursions Variant	0.000000	-	0.938062	+
14	Serial	0.128132	+	0.177628	+
15	Linear Complexity	0.000000	-	0.566688	+

**Table 3 sensors-22-09700-t003:** Results of the investigation of the MAFG repetition period.

*h*	$T_{\mathrm{period}}$	
	a=0	$a=a_{0}⊕a_{1}$
6	$>10^{9}$	$>10^{9}$
7	109,531,200	$>10^{9}$
8	4,193,300	$>10^{9}$
9	193,443,432	$>10^{9}$
10	$>10^{9}$	$>10^{9}$

**Table 4 sensors-22-09700-t004:** Parameters of the combined generator.

*h* Value	LC Parameters	Initial Values
*h* = 4	$a=a_{0}⊕a_{1}$	*Y*_0 = 7, 32, 100 *X*_0 = 7
*h* = 5	$a=a_{0}⊕a_{2}$	
*h* = 6	$a=a_{0}⊕a_{2}$	
*h* = 7	$a=a_{0}⊕a_{2}$	
*h* = 8	$a=a_{0}⊕a_{2}$	
*h* = 9	$a=a_{0}⊕a_{3}$	
*h* = 10	$a=a_{0}⊕a_{3}$	

**Table 5 sensors-22-09700-t005:** *p*-Values of statistical tests for combined generator.

№	Statistical Test	Generator 1		Generator 2		Generator 3	
		*p*-Value	Status	*p*-Value	Status	*p*-Value	Status
1	Frequency	0.943242	+	0.291091	+	0.291091	+
2	Block Frequency	0.044220	+	0.664168	+	0.664168	+
3	Cumulative Sums	0.576961	+	0.950247	+	0.950247	+
4	Runs	0.000000	+	0.377007	+	0.377007	+
5	Longest Run	0.000000	+	0.350485	+	0.350485	+
6	Rank	0.971006	+	0.757790	+	0.757790	+
7	FFT	0.128132	+	0.192724	+	0.192724	+
8	Non-Overlapping Template	0.000000	+	0.626709	+	0.626709	+
9	Overlapping Template	0.000000	+	0.368587	+	0.368587	+
10	Universal	0.000000	+	0.289667	+	0.289667	+
11	Approximate Entropy	0.000000	+	0.096000	+	0.096000	+
12	Random Excursions	0.000000	+	0.958867	+	0.958867	+
13	Random Excursions Variant	0.253205	+	0.938062	+	0.938062	+
14	Serial	0.000000	+	0.177628	+	0.177628	+
15	Linear Complexity	0.653773	+	0.566688	+	0.566688	+

## Data Availability

The data cited in this manuscript are available from the published papers or the corresponding author.

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
