# Peer review of "Combined Pseudo-Random Sequence Generator for Cybersecurity"

_sensors, 2022, doi:10.3390/s22249700_

Round 1
Reviewer 1 Report
In this article, authors have optimized the structures of the classical additive Fibonacci generator and the modified additive Fibonacci generator when they work together. The ranges of initial settings of structural elements (seed) of these generators have been determined, which guarantee acceptable statistical characteristics of the output pseudo-random sequence, which significantly expands the scope of their possible application, including for cybersecurity. As a result of studying the statistical characteristics of the modified additive Fibonacci generator, it was found that they significantly depend on the signal from the output of the logic circuit entered to the structure. It is proved that acceptable statistical characteristics of the modified additive Fibonacci generator, and the combined generator realized on its basis, are provided at odd values of the module of the recurrent equation describing work of such generator. The output signal of the combined generator has acceptable characteristics for a wide range of values of the initial settings of the modified additive Fibonacci generator and the classic additive Fibonacci generator. When it comes to the use of information security, it is worth noting the fact that for modern encryption and security programs, generators of random numbers and bit sequences, and the approaches to their construction, are crucial and critical.
1. Please ensure that all variables/symbols introduced in the manuscript are properly explained and the index of each symbol is correct and consistent in order to avoid confusion.
2. The manuscript is interesting, however, some amendments are required. The related work section should be improved by better discussing the cybersecurity and discussing more related works.
3. Several references about cybersecurity and data analysis must be added [1][2].
[1] Catal, C., Ozcan, A., Donmez, E. et al. Analysis of cyber security knowledge gaps based on cyber security body of knowledge. Educ Inf Technol (2022). https://doi.org/10.1007/s10639-022-11261-8
[2] ÖZCAN, ALPER; ÇATAL, ÇAÄžATAY; TOÄžAY, CENGİZ; TEKİNERDOÄžAN, BEDİR; and DÖNMEZ, EMRAH (2020) "Assessment of environmental factors affecting software reliability: a survey study," Turkish Journal of Electrical Engineering and Computer Sciences: Vol. 28: No. 4, Article 3. https://doi.org/10.3906/elk-1907-49
Author Response
Dear Editors of "Sensors", Section "Sensor Networks" Special Issue "Data Privacy, Security, and Trust in New Technological Trends"
Dear Reviewer,
Taking into account the remarks of respected Reviewers to the manuscript ID: sensors-2047189 “Combined Pseudo-Random Sequence Generator for Cybersecurity” authors: Volodymyr Maksymovych, Mariia Shabatura, Oleh Harasymchuk, Ruslan Shevchuk, Pawel Sawicki, Tomasz Zajac, we have made the following changes to our manuscript:
Reviewer's comments |
Authors' comments |
1. Please ensure that all variables/symbols introduced in the manuscript are properly explained and the index of each symbol is correct and consistent in order to avoid confusion. |
Corrections have been made. |
2. The manuscript is interesting, however, some amendments are required. The related work section should be improved by better discussing the cybersecurity and discussing more related works. |
Information added. |
3. Several references about cybersecurity and data analysis must be added [1][2]. |
Thanks for your advise. We will detail consider suggested by you articles and use them in our next paper. |

Reviewer 2 Report
The authors present an interesting idea in the field of cyber security. It should be agreed that for information security in modern Security programs, the generation of bit sequences and random values ​​is of colossal significance.
The authors in Introduction present the basics that were at the basis of their research and the solution shown. It's good that the authors refer to the quoted sources. After all, I think that this chapter requires transformation. There should be two chapters. One Introduction containing part of this current plus more emphasis on what? why? Why is our solution better?
The second is Related Works, where some of the current one should be moved but plus additionally extended to other works with similar comments.
The authors show well - understandably, their examination. However, the resource and discussion chapter should be divided into two again. The first should be presented with the results with the relevant comments but it is in the Discussion chapter that there should be a discussion on research and the results received, also compared to the results of other teams. Evidently, the authors lack the right structure of the article - e.g. Imrad.
Author Response
Dear Editors of "Sensors", Section "Sensor Networks" Special Issue "Data Privacy, Security, and Trust in New Technological Trends"
Dear Reviewer,
Taking into account the remarks of respected Reviewers to the manuscript ID: sensors-2047189 “Combined Pseudo-Random Sequence Generator for Cybersecurity” authors: Volodymyr Maksymovych, Mariia Shabatura, Oleh Harasymchuk, Ruslan Shevchuk, Pawel Sawicki, Tomasz Zajac, we have made the following changes to our manuscript:
Reviewer's comments |
Authors' comments |
The authors in Introduction present the basics that were at the basis of their research and the solution shown. It's good that the authors refer to the quoted sources. After all, I think that this chapter requires transformation. There should be two chapters. One Introduction containing part of this current plus more emphasis on what? why? Why is our solution better? |
Corrections have been made. We transformed chapter and added table “Comparison of related work with the study made in this paper”, which show benefits our paper.
|
The second is Related Works, where some of the current one should be moved but plus additionally extended to other works with similar comments. |
|
The authors show well - understandably, their examination. However, the resource and discussion chapter should be divided into two again. The first should be presented with the results with the relevant comments but it is in the Discussion chapter that there should be a discussion on research and the results received, also compared to the results of other teams. Evidently, the authors lack the right structure of the article - e.g. Imrad. |
Information was added. Comparison of the results of our investigation with known others were made in the context of the article. That why, we did not create a separate chapter “Discussion”. |

Reviewer 3 Report
The article has a very interesting topic, but some points are not clear.
In the introduction, I strongly suggest that the authors make it clear what novelties this work has in relation to what is already consolidated.
It is imperative to include a comparative table in subsection 1.1. Related Literature. Listing which parameters were considered, and which criteria of the researched works in relation to this one of yours. I suggest at least 7 related and well-clarified works.
This recently published work I believe may contribute to the authors https://doi.org/10.3390/s19194312
In section 2. Materials and Methods
What are the open issues? What problem or problems do the authors propose to solve with the proposed materials and methods?
Nothing is clear...
It is evident in the following section the results obtained do not have sufficient evidence of the metrics used. I suggest, if possible, making available a snippet of algorithms so that the reader can better understand the results obtained.
The comparisons made? reasoning? what it consists of... the definitions used are not clear, it needs to be made clear and clarified.
Figures 4 and 5 are just one line, is it worth a chart in this format? I suggest reviewing...
In the last figure number 10, it is necessary to discuss and explain the graphics better. It is not clear the scientific contributions obtained.
It is evident in the authors' conclusions, it is poor and without evidence of novelties, research problems solved problems, and scientific contributions obtained. Neither are there definitions of future work and discussion of the results obtained that can be improved.
I suggest including a list of mathematical symbols
I suggest including a list of abbreviations
Review references, some are not completely missing DOI or ISSN.
I hope I have contributed to the improvement of the work, I also hope to receive corrections and news soon.
Author Response
Dear Editors of "Sensors", Section "Sensor Networks" Special Issue "Data Privacy, Security, and Trust in New Technological Trends"
Dear Reviewer,
Taking into account the remarks of respected Reviewers to the manuscript ID: sensors-2047189 “Combined Pseudo-Random Sequence Generator for Cybersecurity” authors: Volodymyr Maksymovych, Mariia Shabatura, Oleh Harasymchuk, Ruslan Shevchuk, Pawel Sawicki, Tomasz Zajac, we have made the following changes to our manuscript:
Reviewer's comments |
Authors' comments |
In the introduction, I strongly suggest that the authors make it clear what novelties this work has in relation to what is already consolidated. |
Changes have been made. |
It is imperative to include a comparative table in subsection 1.1. Related Literature. Listing which parameters were considered, and which criteria of the researched works in relation to this one of yours. I suggest at least 7 related and well-clarified works. |
Corrections have been made. Also, we created comparative table, which show benefits of our paper with relative. |
In section 2. Materials and Methods What are the open issues? What problem or problems do the authors propose to solve with the proposed materials and methods? Nothing is clear... |
Changes have been made. |
It is evident in the following section the results obtained do not have sufficient evidence of the metrics used. I suggest, if possible, making available a snippet of algorithms so that the reader can better understand the results obtained. |
As for the algorithms of the combined generator. AFG functions in accordance with of the recurrent equation (3), and the MAFG is in accordance with the recurrent equation (4) and the logical equation (5). In work [42], it was proved that the operation of the MAFG in accordance with equation (4) is ensured by the inclusion of the MUX multiplexer, the second adder AD 2 and logical element OR. In the same work, corresponding algorithms are given that confirm identity of abstract and hardware models. Algorithms used in the hardware model. The MAFG of this work are similar. They differ from the algorithms of the hardware model of the MAFG in work [42] only by the number of registers. In this regard, since the principle of operation of the IAF is not the main one content and novelty of this work, we believe that it is not necessary to cite them. |
The comparisons made? reasoning? what it consists of... the definitions used are not clear, it needs to be made clear and clarified. |
Thank you! It was a translation error. Changes have been made. |
Figures 4 and 5 are just one line, is it worth a chart in this format? I suggest reviewing... |
Fig. 4 and 5 show confirmation of the invariance of the repetition period. It is an important condition for generator characteristics. Therefore, it is necessary to keep these features unchanged. |
In the last figure number 10, it is necessary to discuss and explain the graphics better. It is not clear the scientific contributions obtained. |
Corrections have been made |
It is evident in the authors' conclusions, it is poor and without evidence of novelties, research problems solved problems, and scientific contributions obtained. Neither are there definitions of future work and discussion of the results obtained that can be improved. |
Changes have been made in conclusions. |
I suggest including a list of mathematical symbols |
All mathematical symbols have an explanation. |
I suggest including a list of abbreviations |
All abbreviations have an explanation. |
Review references, some are not completely missing DOI or ISSN. |
DOI and ISSN have been added |

Round 2
Reviewer 2 Report
The authors answered my questions and made corrections.
Author Response
Dear Reviewer,
Thank you for your valuable comments.
To see all the changes in the article, we used track change and also send a cover letter explaining each comment. We accepted all your comments and wishes and reacted accordingly. Please review.
We emphasize that, as stated in the text of the article, the main advantage of the proposed modified MAGF generator is the possibility of implementing its work algorithm based on polynomials with an arbitrary value of the modulus of the recurrent equation and combining this advantage with the advantage of the classical AGF, which together with the MAGF forms a combined generator.
Made it possible to ensure the stability of the repetition period of the output sequence in a wide range of initial settings (key code values). These advantages of this development, as well as considering the fact that we have proposed a hardware implementation of the generator, are what favourably distinguish our work from the known ones (shown in Table 1). It indicates the promising development of further scientific research.
As can be seen from the results of your plagiarism analysis 76 matching positions are less than 1%. The highest similarity value is 9%. It can be seen from the file that most of the matches are based on formal features (surname, designation and explanation of designations, commonly accepted phrases, numbering, author's contribution, etc.). However, there are no matches for scientific novelty and results.
Regards,
the author's team
Reviewer 3 Report
Dear authors, I consider that the changes made according to the previous review were minimal, and the justifications were also not convincing. In addition, according to the Turnitin tool, your text has 42% similarity, that is, almost half is similar to what is already published in other sources. Therefore, I will send the attached report for your correction, if this is not corrected or very well justified, I will have no choice but to reject your article.
Therefore, reviewing what was previously requested, I suggest including an appendix with results and concrete evidence.
Check the similarity of the existing text, according to the attached report.

Author Response

(The authors gave the same response as above.)
